Toxicity of Melaleuca alternifolia essential oil to the mitochondrion and NAD+/NADH dehydrogenase in Tribolium confusum

Liao Min 1 2
Yang Qian-Qian 1
Xiao Jin-Jing 1
Huang Yong 1
Zhou Li-Jun 1
Hua Ri-Mao 2
Cao Hai-Qun 1 2 haiquncao@163.com
1 School of Plant Protection, Anhui Agricultural University , Hefei , China
2 Provincial Key Laboratory for Agri-Food Safety, Anhui Province, Anhui Agricultural University , Hefei , China
Vulpe Chris
Electronic publication date: 2018 Nov 13
Publication date: 2018
Volume: 6
Electronic Location ID: e5693
Received 2018 May 2; Accepted 2018 Sep 5
Copyright: © 2018 Liao et al.
Copyright year: 2018
Copyright holder: Liao et al.
License: This is an open access article distributed under the terms of the Creative Commons Attribution License, which permits unrestricted use, distribution, reproduction and adaptation in any medium and for any purpose provided that it is properly attributed. For attribution, the original author(s), title, publication source (PeerJ) and either DOI or URL of the article must be cited.
License URL: https://creativecommons.org/licenses/by/4.0/

Keywords: Melaleuca alternifolia essential oil, Tribolium confusum, Transcriptome, NAD+/NADH, Transmission electron microscopy

Funding: Anhui University Talent project gxbjZD2016024 Talent Research Project of Anhui Agricultural University yj2018-04 National Forestry Public Welfare Profession Scientific Research Special Project of China 201404601 National Key Research and Development Program of China 2017YFD0201203 This study was supported by the Anhui University Talent project (gxbjZD2016024), Talent Research Project of Anhui Agricultural University (yj2018-04), the National Forestry Public Welfare Profession Scientific Research Special Project of China (NO. 201404601), and National Key Research and Development Program of China (NO. 2017YFD0201203). The funders had no role in study design, data collection and analysis, decision to publish, or preparation of the manuscript.

==============================
Background

In our previous study, Melaleuca alternifolia essential oil (EO) was considered to have an insecticidal effect by acting on the mitochondrial respiratory chain in insects. However, the mode of action is not fully understood.

Methods

In this study, we investigated the insecticidal efficacy of the M. alternifolia EO against another major stored-product pest, Tribolium confusum Jacquelin du Val. Rarefaction and vacuolization of the mitochondrial matrix were evident in oil-fumigated T. confusum adults.

Results

Alterations to the mitochondria confirmed the insecticidal effect of the M. alternifolia EO. Furthermore, comparative transcriptome analysis of T. confusum using RNA-seq indicated that most of the differentially expressed genes were involved in insecticide detoxification and mitochondrial function. The biochemical analysis showed that the intracellular NAD+/NADH ratio is involved in the differential effect of the M. alternifolia EO.

Discussion

These results led us to conclude that NAD+/NADH dehydrogenase may be the prime target site for the M. alternifolia EO in insects, leading to blocking of the mitochondrial respiratory chain.

Introduction

Essential oils (EOs) obtained from aromatic plants by steam distillation are regarded as a new and safe alternative to conventional insecticides because of their bioactive potential and high volatility (Bai et al., 2015; Liang et al., 2016). The volatile nature of plant EOs reduces concerns regarding residues of their constituents on stored grains, which mitigates environmental contamination and effects on non-target organisms (Li et al., 2013; Polatoglu et al., 2016). The potential for synergistic or additive effects between the volatile secondary metabolites of EOs, which act on insects via various targets and mechanisms, is also an advantage (Abdelgaleil et al., 2015).

Essential oils are mixtures of volatile secondary metabolites, mainly monoterpenes and sesquiterpenes; therefore, they have various modes of action in insects. It is difficult to separate and purify an active substance to study its mechanisms. Moreover, the insecticidal activity of an EO can be attributed to the synergistic effects of its major components (Wu et al., 2015); therefore, all the major components need to be considered while assessing the mechanism of an EO. The mechanisms underlying the toxicity of EOs have been explored for decades. To date, most of the studies focused on enzyme inhibition or induction (Ali-Shtayeh et al., 2018; Kiran et al., 2017; Vanhaelen, Haubruge & Francis, 2001). The main targets of EOs are neurotoxic target enzymes such as acetylcholinesterase and a variety of detoxifying enzymes such as glutathione S-transferase (GST) and carboxylesterase (CarE) (Otero et al., 2018). EOs have lethal and sublethal effects on pests that attack stored grains, and they are frequently applied via fumigation by stored-grain managers (Haddi et al., 2015; Silva et al., 2017). Thus, the activity of EOs may cause abnormal respiration (De Carvalho et al., 2017), which is similar to the effect of octopamine (Enan, 2005); however, relatively little is known about the underlying mechanisms, particularly the mitochondrial electron transport chain.

Recently, transcriptome profiling analysis has increased our understanding of insect response to various stressors (Chen et al., 2016; Du, Jin & Ren, 2016). RNA-seq is an effective tool for studying the extensive regulation at transcriptional levels (Clements et al., 2016; Hamisch et al., 2012), and it can be used for characterizing the complexity of mitochondrial transcriptomes (Stone & Storchova, 2015). In our previous study, we had reported, for the first time, a comprehensive transcriptome analysis of the maize weevil, Sitophilus zeamais, to identify the genes and pathways that are likely to be changed upon exposure to the EO obtained from Melaleuca alternifolia (Liao et al., 2016). Our findings suggested that the mitochondrial electron transport chain is a likely target in insects. Such information contributes to new insights on the biological response of insects to EOs and helps us in understanding the molecular mechanisms underlying the insecticidal activity of plant EOs.

M. alternifolia is derived from an Australian plant, M. leucadendron, and it was developed to meet increasing demands for its monoterpene-rich EOs (Bustos-Segura, Külheim & Foley, 2015). Notably, the major component, terpinen-4-ol, showed distinct fumigant toxicity against the pests of stored grains: 6.78 mg/L air of median lethal concentration (LC50) for S. zeamais (Liao et al., 2016). Terpinen-4-ol has also been found in the most EOs reported in previous studies and should be studied further (Abdelgaleil et al., 2015; Brahmi et al., 2016; Yeom et al., 2013).

To obtain information on the fumigant toxicity of M. alternifolia EO and its chemical compounds against stored grain insects and identify a better chemotype, we studied the toxicity of M. alternifolia EO against the confused flour beetle (Tribolium confusum Jacquelin du Val.), which is closely related to the flour beetle T. castaneum (Herbst) (Golestan et al., 2015). To expand on the applicability of our previous transcriptomic analysis and provide a clearer picture of the mode of action of natural insecticides, we also performed RNA-seq analysis of the T. confusum transcriptome to investigate changes in the abundance of mitochondrial transcripts after exposure to the M. alternifolia EO. To verify the reliability of the RNA-seq data, we tested the inhibitory effects of the EO on NAD+/NADH dehydrogenase, which is a possible insecticidal target. Subsequently, we assessed the action of the M. alternifolia EO in degrading the mitochondria in the cells obtained from oil-fumigated T. confusum. To our knowledge, no studies on the molecular events underlying the response of T. confusum to plant EOs have been performed or published.

Materials and Methods

EO and chemicals

The EO (density, 0.8978) was purchased from Fujian Senmeida Biological Technology Co., Ltd (Xiamen, China). Terpinen-4-ol (40.09%), γ-terpinene (21.85%), α-terpinene (11.34%), α-terpineol (6.91%), and 1,8-cineole (1.83%) were the major compounds.

Insect culture

A culture of T. confusum was maintained in the laboratory, and the insects were not exposed to any insecticides. For insect culture, the larvae were reared on sterilized whole wheat at 28 ± 1 and 68 ± 5 °C relative humidity under complete darkness. Then, pupae of the same age were collected and transferred to a new container. After emergence, the adults were reared to about 2 weeks of age for use in the subsequent experiments.

Fumigant toxicity assay

The fumigant toxicity of M. alternifolia EO against T. confusum was determined according to our previous protocol (Liao et al., 2016). For oil exposure, 30 adults were exposed to serial dilution doses in sealed gas-tight 300 mL glass jars and incubated for 24, 48, and 72 h at 28 °C. Drops of the oil (1.8, 2.1, 2.5, 3.2, and 4.0 mL) were applied with an Automatic Micro-applicator (Burkard 900X; Burkard Scientific Ltd., Uxbridge, UK) to a piece of filter paper (2 × 3 cm), and the filter paper was attached to the undersurface of the jar lid. Equivalent groups of control adults were treated similarly, but without exposure to the oil. Three biological replicates were maintained for each treatment. For the EO constituents, the protocol for fumigant toxicity was determined using the above-mentioned process, and serial dilutions were prepared and applied to filter paper. In addition, T. confusum specimens exposed to LC50 (6.37 mg/L air) of oil for 12, 24, 36, 48, 60, and 72 h were collected and washed twice or three times with pre-cooled saline, flash-frozen in liquid nitrogen, and stored at −80 °C for the subsequent bioassays.

Transmission electron microscopy of mitochondria

Cells were obtained from the thorax for transmission electron microscopy (TEM) by dissecting the insects at 24, 48, and 72 h after the oil treatment. The samples were fixed in a mixture of 5% glutaraldehyde and 0.1M sodium cacodylate (pH 7.2) for 24 h. After fixation, the samples were washed, dehydrated, and embedded in pure resin, according to the protocol of Correa et al. (2014). After polymerization in gelatin capsules, ultrathin sections were placed on copper grids and subsequently observed and photographed using a transmission electron microscope (HT7700; Hitachi, Tokyo, Japan).

RNA sequencing

Total RNA was extracted from oil treatment and control groups (collected at 24 h) with TRIzol reagent (Kangwei Century Biotechnology Co. Ltd., Beijing, China), according to the manufacturer’s instructions, and treated with DNase I (Sangon Biotech, Shanghai, China). The RNA quality was checked with a 2100 Bioanalyzer (Agilent Technologies, Santa Clara, CA, USA). Library construction and Illumina sequencing were performed at BGI-Tech (Wuhan, China). For cDNA library construction, five μg of RNA per sample from three biological replicates were combined and used. Two cDNA libraries were constructed for the oil treatment and control groups. For Illumina sequencing, which followed the protocol of the Illumina TruSeq RNA Sample Preparation Kit (BGI-Tech, Wuhan, China), 2 × 100 bp paired-end reads were sequenced using Illumina HiSeq™ 4000 (Illumina Inc., San Diego, CA, USA), with the depth of six g for each sample. The reads were submitted to the NCBI Sequence Read Archive (SRA; accession number, SRS2593554).

Bioinformatic analyses

The reads for the treatment and control groups were mapped to the 165.944-Mb T. castaneum reference genome obtained from NCBI (BioProjects: PRJNA12540) by using TopHat v.2.08 (Kim et al., 2013a), with quality aware alignment algorithms (Bowtie v.2.2.5) (Langmead et al., 2009).

The raw RNA-seq reads were assessed for quality with FastQC (version 0.11.4; Babraham Bioinformatics, Cambridge, UK) and saved as FASTQ files with default parameters (Cock et al., 2010). Then, de novo assembly of the clean reads was performed using the Trinity method (version 2.0.6) (Grabherr et al., 2011). All the unique Trinity contigs were analyzed using BlastX (E-value < 10−5) against the protein databases Nr (Agarwala et al., 2016), Nt (Agarwala et al., 2016), COG (Tatusov et al., 2000), KEGG (Kanehisa & Goto, 2000), Swiss-Prot, and InterPro using InterProScan5 with default parameters. To annotate the assembled sequences with GO terms, Nr Blast results were imported into Blast2GO (Conesa et al., 2005).

Transcript abundance was calculated as fragments per kilobase of transcript per million fragments mapped (FPKM) for each sample (Li & Dewey, 2011). Differential gene expression analysis (fold changes) and related statistical significance in pair-wise comparison were performed using the DESeq program (http://www-huber.embl.de/users/anders/DESeq/) (Anders & Huber, 2010). The differentially expressed genes (DEGs) were identified using a false discovery rate (FDR) threshold ≤ 0.001 and absolute value of log2Ratio ≥ 1 (Hao et al., 2016). Genes with an adjusted p-value were used for controlling FDR, and those with a threshold < 0.05 were classified as differentially expressed (Ma et al., 2015).

For each DEG, GO and KEGG enrichment analyses were conducted using the DESeq R package (http://www.geneontology.org/ and http://www.genome.jp/kegg/, respectively). The GOslim annotations results were then classified into three main classes: molecular function, biological process, and cellular component. The KEGG database was used to identify significantly enriched metabolic pathways or signal transduction pathways.

Quantitative real-time PCR

Quantitative real-time (qRT-PCR) was used to further validate and quantify the RNA levels for 20 selected genes that encode NADH or NAD+ by using the iCycler iQ Real-time Detection System (Bio-Rad, Hercules, CA, USA). Gene-specific primers were designed using Primer Premier 5, and the sequences are listed in Table S1. The house-keeping gene glyceraldehyde 3-phosphate dehydrogenase was used as the reference gene, as proposed by Pan et al. (2015). For the qRT-PCR analysis, cDNA templates were diluted 20-fold in nuclease-free water. Then, mRNA levels were measured in triplicate (technical repeats) with qPCR by using the SYBR Green Master Mix (Vazyme Biotech Co., Ltd, Nanjing, China), according to the manufacturer’s instructions. PCR amplification was performed in a total volume of 20.0 μL containing 10.0 μL of the SYBR Master Mix, 0.4 μL of each primer (10 μM), 2.0 μL of cDNA, and 7.2 μL of RNase-free water. The amplification procedure was composed of an initial denaturation step at 95 °C for 5 min, followed by 40 cycles of 95 °C for 10 s and 60 °C for 30 s and the melting curve step at 95 °C for 15 s, 60 °C for 60 s, and 95 °C for 15 s. Gene expression was quantified (mean ± SD) as relative fold change by using the 2−ΔΔCT method (Schmittgen & Livak, 2008).

Measurement of intracellular NAD+/NADH ratio

Both oxidized and reduced forms of intracellular NAD were determined using an NAD(H) quantification kit (Nanjing Jiancheng Bioengineering Institute, Nanjing, China). Briefly, 0.1 g of the test insects were collected at 12, 24, 36, 48, 60, and 72 h and extracted with one mL of NAD+/NADH extraction buffer in three freeze/thaw cycles. The samples were centrifuged at 10,000×g for 5 min at 4 °C. Then, 0.5 mL of the extracted NADH or NAD+ supernatant was transferred to a centrifuge tube and neutralized with an equal volume of the opposite extraction buffer. The samples were centrifuged at 10,000×g for 10 min at 4 °C and then used for the subsequent bioassays. NADH or NAD+ cycling mix was prepared according to the manufacturer’s protocol (Nanjing Jiancheng Bioengineering Institute, Nanjing, China). Finally, absorbance was measured at 570 nm. In addition, the concentration of the total protein was determined using the total protein quantitative assay (Nanjing Jiancheng Bioengineering Institute, Nanjing, China). Three replicates were used for each treatment, and each replicate was determined three times.

Statistical analysis

The mortality rates observed in the toxicity bioassays were corrected for the control group by using Abbott’s formula (Abbott, 1925). All data are expressed as mean ± SE values of three independent experiments and analyzed using one-way nested analysis of variance and unpaired sample t-test. A significant difference was accepted at a p-value < 0.05. An extremely significant difference was accepted at p-value < 0.01. The LC50 values were evaluated using probit analysis (Hubert & Carter, 1990), and corresponding confidence intervals at 95% probability were obtained using IBM SPSS Statistics 22.0 (IBM, Armonk, NY, USA). Figures depicting the effects of the EO on enzymatic activities and the qRT-PCR results were created using Origin Pro 9.0 (Origin Lab Corporation, Northampton, MA, USA).

Results

Fumigant toxicity of M. alternifolia EO and constituents

To investigate the toxicity of the M. alternifolia EO against T. confusum adults, we performed the fumigation assay. The results show that M. alternifolia EO has potent fumigant toxicity (Fig. 1A), and the effect of fumigation gradually increased over time (24, 48, and 72 h); the corresponding LC50 values were 7.45, 7.09, and 6.37 mg/L air, respectively (Fig. 1B). The largest dose of 11.97 mg/L air EO caused 91.11%, 97.78%, and 98.86% mortality, respectively, in the T. confusum adults.

Figure 1 Fumiganttoxicity of M. alternifolia essential oil (A) and its constituents (C) against T. confusum adults and the corresponding regression analysis (B).

Results are reported as mean ± SE (calculated from three independent experiments). The LC50 values were subjected to probit analysis (Fong et al., 2016). Different lowercase letters at the top of the columns mean significant differences at a p-value of 0.05. The error in Fig. 3C represents the 95% fiducial limits.

In particular, terpinen-4-ol was the most potent toxicant with an LC50 value of 3.83 mg/L air (Fig. 1C). In the M. alternifolia EO, terpinen-4-ol was the main component (40.09% of the EO), indicating that terpinen-4-ol is the major contributor to the fumigant toxicity of the EO. In addition, γ-terpinene and α-terpinene exhibited weaker fumigant toxicity (LC50 = 28.52 and 44.53 mg/L air, respectively) against T. confusum.

TEM of mitochondria

An ultra-structural examination of the morphology of the mitochondria from untreated and oil-fumigated T. confusum larvae is shown in Fig. 2. In the untreated T. confusum larvae, the mitochondria have highly electron-dense cristae, membranes, and matrix (Figs. 2A, 2C and 2E). However, the mitochondria in the columnar and regenerative nidi cells from the thorax of the oil-treated T. confusum larvae had undergone ultra-structural changes detected by the vacuolization of the mitochondrial matrix (Figs. 2B, 2D and 2F), when compared with the non-fumigated adults. The vacuolization increased with time after the oil treatment and, in severe cases, caused fragmentation of the mitochondria.

Figure 2 Ultra-structure of the mitochondria from the thorax of non-fumigated (A, C, and E) and fumigated (B, D, and F) T. confusum adults.

(A) The normal structure of the mitochondrion with many highly electron-dense cristae. (B) A part of the thorax and ultra-structural changes in the mitochondria represented by vacuolization (Vm) and rarefaction (Rm) of the mitochondrial matrix (arrow). (C) The vacuolization aggravated 24 (B), 48 (D), and 72 h (F) after oil treatment. Scale bar = 2.0 μm.

Illumina sequencing and de novo assembly

To obtain a global, comprehensive overview of the T. confusum transcriptome, RNA was extracted from the treatments and control groups. A total of 126,280,032 paired-end reads (100 bp) were generated from the samples by using the Illumina HiSeq™ 4000 platform. Then, 89,342,546 clean reads were obtained by preprocessing and filtering the reads (low-quality sequences were removed; Table 1). Subsequently, the clean reads were subjected to transcriptome assembly by using the Trinity software package (Grabherr et al., 2011), and 28,885 assembled unigenes were generated using overlapping information from high-quality reads, which accounted for 36,998,010 bp (Table 1). Of the assembled unigenes, approximately 38.54% were ≤600 bp and 61.46% were >500 bp. The average length of the unigenes was 1,280 bp, with an N50 length of 2,097 bp and mean length of 1,280 bp. The length distribution of the unigenes is shown in Fig. 3A.

Table 1 Summary of the sequencing reads of the T. confusum transcriptome and corresponding assemblies and statistics of the annotation results.

		Control	Treatment	
Raw reads	Total number	64,760,250	61,519,782	
Total number	45,114,010	44,228,536	
Clean reads	Total nucleotides (nt)	4,511,401,000	4,422,853,600	
Q20 (%)	97.38	97.55	
Contigs	Total number	34,747	33,988	
Mean length (bp)	1,043	1,042	
Primary unigenes	Total number	26,367	25,883	
Final unigenes	Total number	28,885	
Total length (bp)	36,998,010	
Mean length (bp)	1,280	
N50 (bp)	2,097	
GC (%)	37.39	
Number < 600 bp	38.54	
Number ≥ 600 bp	61.46	
Annotation	Nr	23,160 (80.18%)	
Nt	9,941 (34.42%)	
COG	9,451 (32.72%)	
KEGG	18,074 (62.57%)	
GO	6,333 (21.92%)	
Swiss-Prot	18,187 (62.96%)	
InterPro	17,837 (61.75%)	
All databases	23,571 (81.60%)	

Figure 3 Length distribution of assembled sequences (A) and GO (B) and KEGG (C) functional classifications of assembled unigenes of T. confusum.

The reads from four libraries were assembled into 28,885 transcripts.

Functional annotation of T. confusum transcripts

All the assembled unigenes were aligned against seven public databases (Table 1). Of the 28,885 assembled unigenes, 23,160 (80.18%) exhibited sequence similarity to a sequence within the Nr database; 23,571 (81.60%) unigenes were annotated in at least one database, indicating that just a few unigenes (18.40%) could not be identified. The homologous genes that showed the best match (54.24%) were from T. castaneum (91.82%). On the basis of the Nr annotation, GO functional analysis of the unigenes was performed. A total of 6,333 (21.92%) unigenes were assigned to the biological process, molecular function, and cellular component categories, including 57 GO terms (Fig. 3B). In addition, 18,074 (62.57%) unigenes were divided into 42 subcategories and 295 KEGG pathways by using the KEGG annotation system with default parameters to predict the metabolic pathways (Fig. 3C).

Differential expression analysis and pathway enrichment

The sequence analysis and annotation of all the unigenes in S. zeamais fumigated by the M. alternifolia EO provided some valuable information for analyzing the T. confusum transcriptome. From the 23,571 unigenes identified in the analysis, we chose to focus on transcripts encoded by the genes associated with known mechanisms to cope with xenobiotic compounds, including quantitative or qualitative changes in major detoxification enzymes and transporters to decrease exposure (pharmacokinetic mechanisms) or changes in target site sensitivity (pharmacodynamic mechanisms) (Bajda et al., 2015). Specifically, changes in the expression levels of four classes of enzymes and proteins (GST, CarE, cytochrome P450 monooxygenases, and mitochondrial respiratory chain-related proteins) were investigated to determine whether patterns emerged in the upregulation or downregulation of specific transcripts. The transcriptome of T. confusum showed that the largest and most abundant group was ATPase transporters, followed by cytochrome P450s; some of them may be involved in insecticidal mechanisms. The transcriptome also showed five possible NAD+/NADH dehydrogenase transcripts, which may be the main targets for the EO.

For comparison, FPKM of each transcript was calculated to estimate the expression levels between the oil-fumigated and oil-free samples. The important DEGs (999 upregulated and 1,209 downregulated) were identified on the basis of threshold FDR < 0.01 and fold change 2 between the oil-fumigated and oil-free samples. To annotate these DEGs, both GO and KEGG functional analyses were performed.

The GO annotation analysis classified 632 DEGs into three GO categories and 339 terms (Fig. S1A). In the molecular function category, 560 DEGs were classified into 11 terms, namely, antioxidant activity, binding, catalytic activity, electron carrier activity, enzyme regulator activity, guanyl-nucleotide exchange factor activity, molecular transducer activity, nucleic acid binding transcription factor activity, receptor activity, structural molecule activity, and transporter activity.

Among the DEGs, 1,180 unigenes were mapped to 287 different KEGG pathways and five categories (Fig. S1B). According to the threshold of Q-value < 0.05, 22 pathways were significantly enriched (Table S2). Many DEGs were significantly enriched in the metabolism pathways associated with respiration and metabolism of xenobiotics, suggesting that abnormal respiration and metabolic disorders occurred in the T. confusum adults after fumigation with the M. alternifolia EO. In addition, 92 possible insect hormone biosynthesis transcripts, some of which are known targets of chlorbenzuron, were detected (Xu, Xu & Wu, 2017).

To verify the expression patterns of the DEGs involved in metabolism, 20 genes were selected for qRT-PCR analysis. As shown in Fig. 4, similar trends of upregulation/downregulation of the selected DEGs were observed between the qRT-PCR and transcriptome data, confirming the accuracy of our transcriptome profiling.

Figure 4 Real-time qRT-PCR analysis of DEGs that encode respiration and detoxification-related enzymes in T. confusum after oil fumigation.

Gene expression (mean ± SE) was quantified as relative fold change by using the 2−ΔΔCT method. The asterisks indicate significant differences in the expression level of DEGs between the oil-treated and no-oil-treated samples (*p-value < 0.05 and **p-value < 0.01).

NAD+/NADH ratio in T. confusum fumigated with the M. alternifolia EO

On the basis of a previous study on S. zeamais and the above-mentioned results, the NAD+/NADH ratio in T. confusum fumigated with the M. alternifolia EO was measured to investigate whether the EO acts on NAD+/NADH. In the non-fumigated insects, a decrease in NAD+ and NADH levels was observed over the course of 24–48/60 h, which may be affected by starvation. Further, we found that treatment with 6.37 mg/L EO significantly increased NAD+ (Fig. 5A) but decreased NADH (Fig. 5B) levels at 12–48 h, when compared with the non-fumigated samples; however, the opposite trend was observed after 60 h. The ratio of NAD+/NADH in T. confusum from 12 to 60 h after treatment decreased (significantly in 24–48 h) and increased after 60 h, but not effectively (Fig. 5C).

Figure 5 NAD+ (A) and NADH (B) in the control and oil-fumigated T. confusum extracts were quantified.

Optical density at 450 nm was recorded and used to calculate the NADH/NAD+ ratio (C). Values (mean ± SE) are from three independent experiments: (∗) p < 0.05 and (∗∗) p < 0.001 for oil fumigation (LC50 = 6.37 mg/L air) vs control (CK).

Discussion

In this study, similar toxicity patterns were observed for the M. alternifolia EO and major compounds (Liao et al., 2016), which confirms this EO as a possible alternative to the natural fumigants currently in use. α-terpinene and γ-terpinene possessed weaker fumigant toxicity against T. confusum than terpinen-4-ol and α-terpineol, showing that the oxygen-containing compounds could cause a remarkable change in bioactivity. Kim, Kang & Park (2013b) described a similar structure–activity relationship among oil constituents with aldehyde, ketone, and alcohol groups and hydrocarbons against rice weevil adults. Terpinen-4-ol and α-terpineol have similar fumigant toxicity, which was more toxic than the EO. The two constituents accounted for 46% of the EO content, and about twofold LC50 of M. alternifolia EO. Thus, the fumigant toxicity of the M. alternifolia EO may be attributable to a synergistic effect of the activities of the oil constituents. We deduced that the terpinen-4-ol chemotype is the main insecticidal active component, which accounted for 40% of the EO content. The amount of terpinen-4-ol directly affects the insecticidal activity of the EO, according to the fumigant toxicities of the constituents of the EO. Terpinen-4-ol is also found in many reported EOs (Du et al., 2014; Liang et al., 2017). Thus, we suggest that the chemotypes of oils rich in terpinen-4-ol should be explored as potential natural insecticides.

Essential oils have produced remarkable results; however, several barriers stand in the way of their application in agriculture. Their unclear mode of action is one of the most significant barriers. An EO is a well-known mixture of volatile secondary metabolites that operate via several modes of action. In insects, octopamine (Enan, 2005) and GABA receptor (Enan, 2001) are considered targets for EO activity. In our previous study, the M. alternifolia EO was suggested to have sub-lethal behavioral effects on insects by blocking the mitochondrial electron transport chain. Inouye et al. (1998) also showed the respiration-inhibitory effects of EOs on filamentous fungi. Similarly, modification of the mitochondria confirmed that fumigation with the M. alternifolia EO affected the mitochondria in the thorax, where the mitochondria became enlarged and swollen. This led to respiratory failure and energy deficiency in the insect body. The results were consistent with those obtained in a previous study in which allyl isothiocyanate oil and PH3 were used (Mansour et al., 2012). Prates et al. (1998) reported that terpenoids had lethal effects on rice weevils because they affected the respiratory and digestive systems. The main components of the M. alternifolia EO are terpinen-4-ol (40.09%), followed by γ-terpinene (21.85%), α-terpinene (11.34%), α-terpineol (6.91%), and α-pinene (5.86%), which are all terpene compounds. The findings of this study are also supported by the morphological alterations, represented by matrix rarefaction and vacuolization, observed in the mitochondria.

However, insecticidal poisoning may occur by affecting different metabolic targets. A previous study has reported that terpenes are very important components of EOs and prone to in vivo metabolism by GST, CarE, and P450s in the insect body (Patra, Das & Baek, 2015). Miyazawa & Kumagae (2001) and Haigou & Miyazawa (2012) also showed that terpinen-4-ol was prone to in vivo metabolism. In our study, the T. confusum transcriptome revealed 54 transcripts that encode cytochrome P450s, with 18 differentially expressed more than twofold and 33 significantly increased (p < 0.05) under oil exposure (Table S3). These genes mainly belong to the CYP6 family. This might explain why terpinen-4-ol can be metabolized by P450s (Haigou & Miyazawa, 2012). Most of the genes that encode CarEs and GSTs were also significantly downregulated upon oil exposure (Table S3). The redundant components may bind to the site of the enzyme, resulting in disturbance of the activity. When the conjugated xenobiotics are translated into innocuous substances, the bound enzymes are damaged. This result is consistent with that observed in a previous study. Overall, T. confusum probably uses these enzymes in combination to catalyze and improve the transformation and degradation of exogenous compounds, resulting in the enhancement of the immune system of the insect. Silencing the upregulated gene expression may contribute to increasing the insecticidal activities of the EO.

Interestingly, we found that five transcripts encoding the subunits of NAD+/NADH dehydrogenase in complex I were significantly upregulated (Table S4). Our biochemical analysis showed that the M. alternifolia EO caused pronounced inhibition of NADH but increased NAD+ level from 12 to 60 h and then subsequently inhibited it. Complex I is the gatekeeper of the respiratory chain and catalyzes the first step of NADH oxidation. NAD+ is a biological oxidizing agent in many metabolic reactions, and tNOX oxidizes hydroquinones and NADH, converting the latter to the oxidized NAD+ form (Titov et al., 2016). It elevates the NAD+/NADH ratio and translocates protons across the inner mitochondrial membrane, which ultimately leads to energy production. To increase energy production in response to oil interference, T. confusum probably coverts NADH excessively to the oxidized NAD+, resulting in an increase in NAD+ levels. However, the regulatory mechanism of T. confusum is destroyed with time, resulting in a significant reduction in NAD+ levels. Therefore, T. confusum recovers the activity of NADH by upregulating the expression of NADH genes; however, the NADH levels have been reduced because of excessive conversion. This might explain our observation that the transcripts encoding the subunits of NAD+/NADH dehydrogenase were significantly upregulated at 24 h.

As reported by De Carvalho et al. (2017), Eugenia uniflora L. (family, Myrtaceae) EO can inhibit the respiratory electron transfer system established with an uncoupler. Parastoo et al. also found that Tagetes minuta EO significantly reduced NADH oxidase (Karimian, Kavoosi & Amirghofran, 2014). Terpenes are the main constituents of the above-mentioned EOs and appears to play an important role in the cellular bioenergetic failure. Moreover, the M. alternifolia EO was observed to alter the morphology and ultrastructure of mitochondria in Botrytis cinerea, which causes mitochondrial dysfunction and disrupts the TCA cycle (Li et al., 2017). Thus, we concluded that NAD+/NADH dehydrogenase may be the prime target for the M. alternifolia EO in insects, leading to blocking of the mitochondrial respiratory chain. This results in a dysfunctional energy system, damage to the mitochondria, and death.

Conclusions

To clarify the applicability of the findings of our previous study, we investigated the action of the M. alternifolia EO in degrading the mitochondria of T. confusum. Alterations to the mitochondria confirmed the insecticidal effect of the M. alternifolia EO, which may act by damaging the mitochondria. To better understand the insecticidal mechanism of the M. alternifolia EO, comparative transcriptome analysis of T. confusum using RNA-seq yielded a total of 2,208 DEGs in response to oil fumigation. The biochemical analysis showed that the intracellular NAD+/NADH ratio is involved in the differential effect of the M. alternifolia EO. Thus, NAD+/NADH dehydrogenase appears to be a prime target for pest control.

Supplemental Information

Supplemental Information 1 Fig. S1. GO (a) and KEGG (b) pathway analysis of DEGs of S. zeamais after oil- fumigation.

Click here for additional data file.

Supplemental Information 2 Table S1. qRT-PCR was used to further validate and quantify the RNA levels for 20 selected genes that encode NADH or NAD+.

qPCR primers and primer efficiency.

Click here for additional data file.

Supplemental Information 3 Table S2. Top 22 enriched KEGG pathways between oil-fumigated and control samples.

Click here for additional data file.

Supplemental Information 4 Table S3. Genes associated with mitochondrial functions were differentially expressed.

Differentially expressed genes in respiration- related enzymes.

Click here for additional data file.

Supplemental Information 5 Table S4. T. confusum transcriptome revealed 54 transcripts that encode cytochrome P450s, with 18 differentially expressed more than 2-fold and 33 significantly increased (p < 0.05) under oil exposure.

Differentially expressed genes in xenobiotic detoxification- related enzymes.

Click here for additional data file.

Supplemental Information 6 Raw data per replicate of partial NAD+/ NADH level and corresponding calculative process.

Click here for additional data file.

Supplemental Information 7 Raw sequence data.

Click here for additional data file.

Additional Information and Declarations

Competing Interests

Author Contributions

Data Availability

The authors declare that they have no competing interests.

Min Liao performed the experiments, prepared figures and/or tables, approved the final draft.

Qian-Qian Yang performed the experiments, analyzed the data, prepared figures and/or tables, approved the final draft.

Jin-Jing Xiao analyzed the data, authored or reviewed drafts of the paper, approved the final draft.

Yong Huang analyzed the data, authored or reviewed drafts of the paper, approved the final draft.

Li-Jun Zhou contributed reagents/materials/analysis tools, approved the final draft.

Ri-Mao Hua contributed reagents/materials/analysis tools, authored or reviewed drafts of the paper, approved the final draft.

Hai-Qun Cao conceived and designed the experiments, authored or reviewed drafts of the paper, approved the final draft.

The following information was supplied regarding data availability:

The reads were submitted to the NCBI Sequence Read Archive (SRA; accession number, SRS2593554).

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
