# Peer review of "Toxicity of Melaleuca alternifolia essential oil to the mitochondrion and NAD+/NADH dehydrogenase in Tribolium confusum"

_PeerJ, doi:10.7717/peerj.5693_

## Round 0.1 · original submission · Major Revisions

Please address the reviewers concerns

Reviewer 1 ·

Basic reporting

The manuscript deals with the potential discovery of the essential oil (EO) modes of action in insects. However, the literature references do mirrow the work done by the authors and not necessarily the classical references with robust background to support the author´s ideas!

There are also several efforts to be applied in the figures in order to increase the readers comprehension. For instance, the panel A in figure 1 is overloaded with unneeded informations and panel B in this figure contains innapropiated (i.e., green and red) collors for reades that suffer of daltonism (or collor blindness). The same patters can also be detected in Figure 4 (information overload) and Figure 5 (collor blindness).

Experimental design

It seems like that these manuscript is a kind of follow up for the authors previous experiments. They only change the insect especies and conducted similar experiments done with other insects. They left it clearly in the first sentence in the abstract.

Validity of the findings

The findings described are indeed relevant and will be of great help for the PeerJ readers. However, the authors needs to reorganize the sifnificance level of their findings. For instance, the discussion sentence in their abstract is not not totally proved in their experiments. I mean, their results do not allow them to conclude that NAD+/NADH dehydrogenase may be the prime target site of ALL essential oils in insects. This conclusion would only be possible if they had conducted such experiments with much more insect species and with definitively many other essetial oil types.

For instance, the sentece in the lines 288-290 might not be completely true if someone use the concerns raised in Robetson et al. 2007 (Robertson JL, Savin NE, Russell RM, Preisler HK (2007) Bioassays with Arthropods, Second Edition -, Second edition. Taylor & Francis Group, Boca Raton, FL). To me, these insects' susceptibilities to this EO is the same! The authors need to show the Susceptibility Ratio (SR, described by Robertson et al. 2007) in order to write something like that!

Reviewer 2 ·

Basic reporting

The text will greatly benefit from language editing, The discussed ideas and findings need to be clearly and concisely reported. Prior Literature needs to be appropritely referenced.

Experimental design

In order to be reproducible, details are still lacking in the method section

Validity of the findings

The findings and the data should be better analyzed and argumented to support the conclusions.

Additional comments

The ms is one of a serie of papers by the same group dealing with the essential oils and their major components ( especifically Melaleuca alternifolia and terpinen-4-ol) and their mode of action in insects.
The authors used a range of methods including toxicological bioassays, microscopy and omics to investigate the insecticidal effects of EOs.
Altough of actual importance, the ms suffers from many flaws and needs a lot of improvements to be published. I have conserns about the methodology, the discussion and the interpretation of the findings.
Below are some comments to be considered by the authros:

Introduction

Lines 56-61: from “EO” to “essential oil” , this section could be moved to the start of the paragraph in the line 43.
Line 74: delete “Progeny of the maize weevil, Sitophilus zeamais, is affected by parental exposure to clove and cinnamon essential oils” and correct the reference it seems duplicated!!!
Lines 78-79: what do authors mean with “while focusing on the standardization of natural insecticides”?? In this ms no standardization was performed!!!!.. Rephrase…

Materials and Methods

Lines 98-99 : please provide more details on how insects of the same age were separated.
Lines 100-110: The toxicity bioassay is not well described… how many concentrations were used to determine the LC50? And how these concentrations were determined? Did the authors use the same protocol to evaluate the toxicity of single components of the EO described in figure 1c?
Line 118: delete “ was”
RNA sequencing: what were the depth / coverage of the sequencing?
Line 133: delete [33] !!
Line 134: delete “.” before the reference.
Line 135: explain the FPKM at first citation!!!
Line 137: the DOI seems to be invalid.. Please check it out!!!
Line 141: correct to “differentially”

In the result section, there are results of the transmission electron microscopy but no corresponding protocol is described in the M&M section!!!

Line 180: the reference is not the best to use!!! … Please use the original reference describing the Probit analysis for LCs determination!!!

Results

Throughout the results section, the authors have been discussing various findings instead of only reporting them!!! See lines 187; 194; 200-204; 212-213; 276-281…

Lines 192- 193 delete from “For the same…
Line 198: invert the order of the α-terpinene and γ-terpinene to correspond to the LC50s!!!
Lines 226-230: From “ namely.. to database”, this part is only repeating what was described in the M&M section!!! Better delete..

Discussion

- The ultra-structural alteration in the mitochondria should be taken carefully, firstly because the authors did not indicate the time after exposure of the sample used and secondly these alterations could be a result of the direct action of the EO on the respiratory process as well as subsequent result of the death process caused by the action of the EO on different metabolic targets. Furthermore, the authors do not discuss or argument over the changes in the NAD + and NADH found in the control along the 72 hrs of evaluation. In addition, the results of the RNAseq showed that beside the genes related to the mitochondrial respiratory chain, over 2000 genes have been found to be differentially expressed including various genes related to xenobiotic detoxification processes!!! Taking in consideration that the sequencing was done 72 hours after the exposure, it is quite difficult to conclude that the alterations found are directly linked with the action of the EO on the mitochondrial respiratory chain ( specifically and solely). On the light of the above, the hypothesis that NAD+/NADH dehydrogenase is the prime target site of EOs in insects still needs stronger arguments and thorough discussion.

- The authors did not discuss the fact that two main components: Terpinen-4-ol and α-terpineol were more toxic (smaller LC50s) than the essential oil itself.

Line 289: correct to 24 h

References

- All the scientific names have to be in italic
- Some references are duplicated: see Li et al., 2013 and Liao et al.,2016.
- Lines 45- 47: references needed to support your
- Line 47-49: the following references may be included
Silva et al., 2017 : “Progeny of the maize weevil, Sitophilus zeamais, is affected by parental exposure to clove and cinnamon essential oils”
Haddi et al., 2015 : “Sublethal exposure to clove and cinnamon essential oils induces hormetic-like responses and disturbs behavioral and respiratory responses in Sitophilus zeamais”

---

## Round 0.2 · Minor Revisions

Please address the minor issues that were raised about the figure presentation

Reviewer 1 ·

Basic reporting

That manuscript increase the scientific quality with the efforts done by the authors!

Experimental design

thais OK now!

Validity of the findings

it is fine and better adjusted now!

Additional comments

I would only strongly recommend some more efforts on the figures. Please, organize the figures 1, 3 and 5 in away that they can increase the font size of the "X" and "Y" axis. Maybe the authors may try to organize these multipanel figures in the vertical stile (suggestion: try to create the figures in a way that they will not exceed the 8.9 cm wide)

Reviewer 2 ·

Basic reporting

the structure of the ms improved and is meeting the standard of PeerJ .

Experimental design

The experimental design is suitable and allow to collect valid results.

Validity of the findings

Findings are sounds and support the conclusions made.

Additional comments

The authors took in consideration all the comments from the previous round of reviewing and the quality of their manuscript dramatically improved.
I do beleive that the findings here presented will be of great interest of the PeerJ readers. Hence I recomend the publication of this manuscript in its actual from.

---

## Round 0.3 · accepted · Accept

The figures are much improved but still minor issue with Figure 5 and cutting off of a plus sign that you can resolve during production.

#